# Imaging of Pancreatic Neuroendocrine Neoplasms

**DOI:** 10.3390/ijerph18178895

**Published:** 2021-08-24

**Authors:** Giuditta Chiti, Giulia Grazzini, Diletta Cozzi, Ginevra Danti, Benedetta Matteuzzi, Vincenza Granata, Silvia Pradella, Laura Recchia, Luca Brunese, Vittorio Miele

**Affiliations:** 1Department of Radiology, Azienda Ospedaliero-Universitaria Careggi, Largo Brambilla 3, 50134 Florence, Italy; giudittachiti@gmail.com (G.C.); dilettacozzi@gmail.com (D.C.); ginevra.danti@gmail.com (G.D.); matteuzzibenedetta@gmail.com (B.M.); pradella3@yahoo.it (S.P.); vmiele@sirm.org (V.M.); 2The Italian Society of Medical and Interventional Radiology (SIRM), SIRM Foundation, 20122 Milan, Italy; 3Radiology Division, Istituto Nazionale Tumori IRCCS Fondazione Pascale—IRCCS di Napoli, 80131 Naples, Italy; v.granata@istitutotumori.na.it; 4Department of Medicine and Health Sciences, University of Molise, 86100 Campobasso, Italy; laura.recchia@unimol.it (L.R.); luca.brunese@unimol.it (L.B.)

**Keywords:** abdominal radiology, gastrointestinal radiology, pancreatic neuroendocrine neoplasms (panNENs), computed tomography (CT), magnetic resonance imaging (MRI), somatostatin receptor scintigraphy (SRS), positron emission tomography (PET)

## Abstract

Pancreatic neuroendocrine neoplasms (panNENs) represent the second most common pancreatic tumors. They are a heterogeneous group of neoplasms with varying clinical expression and biological behavior, from indolent to aggressive ones. PanNENs can be functioning or non-functioning in accordance with their ability or not to produce metabolically active hormones. They are histopathologically classified according to the 2017 World Health Organization (WHO) classification system. Although the final diagnosis of neuroendocrine tumor relies on histologic examination of biopsy or surgical specimens, both morphologic and functional imaging are crucial for patient care. Morphologic imaging with ultrasonography (US), computed tomography (CT) and magnetic resonance imaging (MRI) is used for initial evaluation and staging of disease, as well as surveillance and therapy monitoring. Functional imaging techniques with somatostatin receptor scintigraphy (SRS) and positron emission tomography (PET) are used for functional and metabolic assessment that is helpful for therapy management and post-therapeutic re-staging. This article reviews the morphological and functional imaging modalities now available and the imaging features of panNENs. Finally, future imaging challenges, such as radiomics analysis, are illustrated.

## 1. Introduction

Pancreatic neuroendocrine neoplasms (panNENs) arise from pluripotent cells within the exocrine pancreas and share the common morphologic neuroendocrine differentiation and expression of neuroendocrine markers, such as synaptophysin and chromogranin [1].

Though rare, this group of tumors represents the second most common pancreatic neoplasm, with an estimated annual incidence of 1 per 100,000 individuals; in recent years, this value has been increasing largely due to the improvement of available diagnostic modalities [2]. Besides the sporadic form, about 10–20% of panNENs occur in genetic syndromes, such as multiple endocrine neoplasia (MEN-I), Von Hippel Lindau disease, tuberous sclerosis and neurofibromatosis [3].

Clinically, panNEN can be distinguished into functioning and non-functioning based on the association with a specific syndrome related to hormone overproduction [4]. About 30% of panNENs are functioning type and are diagnosed earlier than non-functioning tumors; insulinomas and gastrinomas are the most common functioning ones [5]. Because of the non-specific clinical symptoms, nonfunctioning tumors present when they exhibit mass effect symptoms or when the disease is already metastatic [6].

The grading system is based on 2017 World Health Organization (WHO) classification, which stratifies panNENs according to cellular differentiation and cellular proliferation. Morphologically, they are distinguished in well-differentiated pancreatic neuroendocrine tumors (PanNETs), poorly differentiated pancreatic neuroendocrine carcinomas (panNECs) and mixed neuroendocrine/non-neuroendocrine neoplasms (MiNENs) of the pancreas. Well-differentiated panNETs typically have an organoid architecture, and according to the Ki-67 proliferation index and/or mitotic index, they are classified into three grades; the Ki-67 cutoff for panNET G1 and G2 is 3% (mitotic rate cutoff is 2/10 high-power fields), while panNET G3 is identified by a high proliferation index (Ki-67 > 20% and/or mitotic rate > 20/10 high-power fields). PanNET G3 and panNEC share the same proliferation system and are distinguished only by cellular differentiation. All of these entities differ significantly in clinical presentations, imaging features, prognosis and management [7,8]. The great update of the 2017 WHO classification is the concept of high-grade (G3) well-differentiated NETs, so that G3 is no longer reserved only for poorly differentiated NECs. The fifth edition of the WHO classification of digestive-system tumors published in 2019 extended the 2017 WHO classification of panNENs for NENs throughout the gastrointestinal tract [9].

Tumor staging can be assessed either by the American Joint Committee on Cancer (AJCC) or by the European Neuroendocrine Tumor Society (ENETS) systems. The most recent classification is the revised AJCC eighth staging system (2017), which introduced the classification criteria, asserted by ENETS. This staging system is used only for G1, G2 and G3 panNETs; indeed, neuroendocrine carcinoma follows the exocrine pancreatic cancer staging system. The T parameter depends on the size of the tumor, and soft-tissue invasion is no more staging criteria. Duodenum or bile-duct invasion is categorized as T3 category; instead, invasion into adjacent structures, including the spleen, stomach, colon, adrenal gland and major vessels, suggests assignment of stage T4. The M1 category has been subdivided into three stages according to hepatic or extra-hepatic localization [10,11].

Although a panNEN diagnosis of certainty is determined by histologic examination of biopsy or surgical specimens, morphologic and functional imaging techniques are crucial for detection, staging, prognosis and determination of treatment strategy [12]. In addition, most of NENs are non-functioning incidentally detected tumors, so it is important for radiologists to suspect NEN on the basis of the imaging features and to be aware of atypical presentation. Furthermore, some NECs can show negative immunohistochemistry marker, and the radiologist, as part of the multidisciplinary team, may suggest the neuroendocrine nature of the lesion [13].

In this setting, many works in the literature describe the crucial role of imaging for patient care. Dushyant et al. provided a panoramic review on diagnosis of gastroenteropancreatic neuroendocrine tumors, with a special emphasis on the role of imaging in tumor detection and characterization [12]. Dromain et al. highlighted the need for a multidisciplinary approach to diagnosis, characterization and staging of panNENs [14]. The purpose of this review is to comprehensively review the role and the recent updates in both morphological and functional imaging modalities describing typical and atypical imaging features of panNENs.

## 2. Morphologic Imaging

Cross-sectional radiological examinations, including computed tomography (CT) and magnetic resonance imaging (MRI), are essential for the localization and staging of panNENs. Therefore, at least one high-quality imaging examination with contrast enhancement is mandatory. Total-body contrast-enhanced CT with a two-phase protocol is useful for primary NEN diagnosis, staging, surveillance and therapy monitoring; however, it shows a low sensitivity for bone metastases. Contrast-enhanced MRI including diffusion-weighted imaging is preferred for examination of the liver and pancreas, but its role in disease staging is limited by the long acquisition time. Ultrasonography (US) is excellent for the initial diagnosis of liver metastases and is the method of choice to guide the biopsy needle for the histopathological NEN grading; endoscopic US is the most sensitive method to diagnose pancreatic NENs [15,16].

### 2.1. Ultrasonography

The use of ultrasonography imaging is almost exclusively limited to Endoscopic Ultra-Sound (EUS). In fact, due to its low sensitivity and specificity, the usefulness of transabdominal ultrasonography is limited to the detection of possible liver metastases [17].

EUS, on the other hand, plays a crucial role when panNEN is suspected especially in case of small size lesion (2 cm), such as insulinomas and gastrinomas, with a sensitivity ranging from 57 to 94% [18,19]. According to ENETS Consensus guidelines, it is considered as the imaging study of choice to be performed when other non-invasive imaging studies have failed in diagnosis [18].

As recently demonstrated by Manta et al., Computed Tomography (CT) failed to detect the lesion in more than 68% of p-NETs with a diameter less than 10 mm, and in a further 15% of patients with a lesion diameter between 11 and 20 mm [20]. In a recent systematic review and meta-analysis, was observed that preoperative EUS evaluation consistently increased the detection of PNETs by over 25% after CT scan [21].

At EUS panNENs usually presents as homogeneously hypoechoic mass with sharp, well-defined margins. Rarely, they may present cystic spaces [22].

Although most pancreatic solid lesions are hypoechoic on EUS; therefore, contrast-enhanced harmonic EUS (CH-EUS) has proven to be useful in differential diagnosis of pancreatic lesions showing a typical rapid and intense enhancement in the early arterial phase, as demonstrated by Kitano et al., who reported this enhancing pattern in 78% of panNENs [23].

EUS can also provide high-resolution pre-operative staging, evaluating the vascular invasion and the distance from the Wirsung duct, driving the decision on the optimal treatment strategy and surgical approach to undertake [24].

However, the most important application of endoscopy is the collection of biopsy sample via Fine Needle Aspiration (FNA), which is crucial for diagnosis and grading of pancreatic tumors [25].

A 2019 retrospective study by Di Leo et al. reported an adequacy of 98.3% in the diagnosis of panNET for the combination of cytology and histology obtained from EUS biopsy with a concordance of 84% between surgery and EUS biopsy comparing the Ki-67 index and grading [26].

Finally, as demonstrated in a cross-sectional study, EUS has greater sensitivity than other diagnostic procedures in identifying pancreatic lesion in patients with a proven MEN-1 mutation or with a MEN-1 manifestation and a mutation carrier in a first-degree family member (detection rate of 94% versus 22% and 12% with CT/MRI or SRS, respectively) [26], suggesting a possible role in the follow-up of patients with MEN-1.

### 2.2. Computed Tomography (CT)

CT is widely used as first line anatomical imaging of choice for initial tumor detection and staging, thanks to its availability, high spatial resolution and rapid acquisition. A number of studies carried out on the diagnostic accuracy of CT in panNENs have showed lesion detection rates ranges between 69 and 94% depending on tumor size and vascularization [21].

Multiphase CT protocol should include non-enhanced CT images in order to identify calcifications or hemorrhage and perform post-contrast acquisition, including an arterial (25–30 s) or pancreatic phase (40–45 s) and a portal venous phase (70–80 s). Due to their hypervascularization, arterial phase, with a sensitivity of 83–88%, shows a clear advantage over venous phase (sensitivity 11–76%), especially in small tumors, such as insulinomas [21,27].

At pre-contrast CT examination, panNENs typically appear as well-rounded, isodense or slightly hypo-dense lesions, with a homogeneous pattern and regular margins [28]. Following the administration of intravenous contrast material, panNENs typically tend to hyper-enhance during arterial phase and demonstrate a slow washout, appearing hyper-dense or isodense among the surrounding parenchyma during portal phase (Figure 1) [29,30]. This typical presentation is more frequently seen in functioning tumors; non-functioning panNENs, especially large ones, more often tend to have an atypical appearance.

The most common atypical presentations are the hypo-enhancing pattern (Figure 2), the intra-vessels growth, the intra-ductal growth and the cystic and calcified variants, aspects that can make difficult the radiological differential diagnosis with pancreatic ductal adenocarcinoma [31]. Furthermore, as recently demonstrate by Kim. Et al., panNENs with uncommon findings are associated with a significantly worse survival rate [31].

Analysis of pancreatic masses solely based on enhancement patterns can be misleading, as panNENs sometimes can have a highly fibrotic stroma and show slow and progressive wash-in. In a recent retrospective study by Jeon et al. on a population of patients with surgically confirmed panNETs, 49% showed a non-hypervascular pattern [32]. On the other hand, the main pancreatic ductal involvement, presenting with upstream duct dilatation, is infrequent, and is an independent predictor for adenocarcinoma [31]. However, in a recent study focused on the identification of CT features distinguishing panNECs from panNETs by Park HJ et al., the first ones demonstrated significantly higher frequencies of main pancreatic ductal dilatation (Figure 3) [33].

Analyzing the CT features of functioning versus non-functioning panNENs, we see that the first finding to describe is that functioning panNENs are often of small size, while non-functioning panNENs are often large; thus, that the challenge of imaging is to determine their extent and the potential of resectability.

Among functioning panNENs the most common ones are Insulinomas (40% of all functioning tumors). They can be located all over the pancreas and generally isolate. Insulinomas show a round shape with a diameter < 2 cm and well-defined borders [32]. As confirmed by the study of Fidler et al., on CT images, these lesions are homogeneously hyper-dense in the arterial/pancreatic phase [34]. The second most common functioning panNEN is Gastrinoma that tend to be less vascular and with a higher probability to be extra-pancreatic than insulinoma. Gastrinomas are small tumors (0.3–3 cm) too, and 80–90% are located within the “gastrinoma triangle”, defined as the space marked by the junction point between the cystic duct with the common hepatic duct, by the second and the third parts of the duodenum and by the connection between the neck and the body of the pancreas. They are often multi-centric, especially when they are associated with MEN1. On contrast-enhanced CT images, gastrinomas show often a delayed enhancement due to presence of fibrosis [34]. The others less common subtype of functioning panNENs (5%) are glucagonoma, somatostatinoma, VIPoma, ACTHoma and PPoma. Usually, they are isolate lesions with heterogeneous enhancement pattern due to necrotic or hemorrhagic aspects or cystic and calcified pattern [32].

On the other hand, non-functioning tumors are 60–80% of panNENs. They tend to be a pancreatic mass with a high rate of tumoral vein thrombosis even more than pancreatic adenocarcinoma, while dilatation of the main pancreatic duct and common bile duct is less common than in pancreatic adenocarcinoma. Their pattern of enhancement is often heterogeneous due to necrotic and hemorrhagic changes [34].

When a hypervascular pancreatic or peri-pancreatic lesion is found, panNENs differential diagnosis should also include metastasis from primary hypervascular malignancies, such as renal cell carcinoma, thyroid cancer, and melanoma, as well as an intra-pancreatic or peri-pancreatic accessory spleen and duodenal gastrointestinal stromal tumors (GISTs) arising from the second portion of the duodenum [33,34,35,36].

CT is mandatory for disease staging with a sensitivity for lymph node metastases from gastro-entero-pancreatic Neuroendocrine Neoplasms (gepNENs) ranging between 60 and 70% and a specificity ranging from 87 to 100% [21,27,37].

The most common sites of metastases are liver, peritoneum, bone and more rarely lungs [38]. Hepatic and lymph node metastases can appear both hyper- or hypo-vascular during arterial phase (Figure 2) [39].

### 2.3. Magnetic Resonance Imaging (MRI)

MR imaging is a suitable alternative to CT in detection and characterization of lesions and is an excellent modality when CT findings are equivocal or inconclusive, thanks to its superior soft-tissue resolution without ionizing radiation.

A sensitivity of 79% (range: 54–100%) and specificity range of 78–100% [26] for MRI in panNEN detection were calculated, with increasing values for tumors greater than 2.5 cm [40]. Moreover, a retrospective study by Farchione et al., observed that MRI with morphological sequences and diffusion-weighted imaging (DWI), and 68Ga-DOTANOC PET/CT had comparable diagnostic results, confirming them as alternative tools [41].

After obtaining initial localizer sequences, MRI protocol should include triplanar (axial, coronal and sagittal) T2-weighted Turbo Spin Echo (TSE) sequences, fat saturation T2 weighted images and T1-weighted in-phase and out-of- phase axial chemical shift imaging, primarily to assess anatomy [21,42]. Axial unenhanced and contrast-enhanced T1W fat-saturated 3D volumetric (GRE) sequences are essential as well. Dynamic contrast enhancement sequences include arterial, venous, and delayed phase 3–5 min after injection start [26].

PanNEN appears, on MRI, as a T1 hypo-isointense lesion within the surrounding hyperintense pancreatic parenchyma, T2 hyperintense or less hypointense and, similarly to CT, hyperenhance during arterial phase with a slow washout (Figure 4) [25]; type of enhancement depends on necrotic and hemorrhagic changes.

On MRI, insulinomas show a high signal on T2-weighted imaging, so that they are often better depicted on this sequence, especially with fat suppression than on T1 weighted in the arterial phase [34]. Gastrinomas are hyperintense on T2W images too. However, gastrinomas usually have a ring-enhancement on contrast-enhanced T1weighted imaging, as Semelka et al. found in their study on 22 panNENs. In addition, in their study, they found that the others functioning panNENs usually enhance heterogeneously [43]. Non-functioning panNENs, in contrast to pancreatic adenocarcinoma, show a high signal on T2-weighted imaging and a vivid contrast enhancement during the arterial/pancreatic phase of the dynamic study. The enhancement can be homogeneous or heterogeneous, and ring- or target-like [32].

The use of MR cholangiopancreatography (MRCP) can be useful in case of pancreatic duct or main bile duct involvement [21,44].

The use of DWI is increasing in the evaluation of panNENs, which show a clearly restrictive pattern; echo planar diffusion-weighted axial imaging with b-values of 50, 500 and 1000 (with calculated apparent diffusion coefficient (ADC) map) are performed to obtain information about tumor cell density. A lot of studies underlined how DWI is helpful to detect and localize especially small panNENs as insulinomas, thanks to its greater image contrast and functional information [45,46,47]. DWI is particularly useful in those patients with contraindications to contrast medium injection and the association of DWI and T2-weighted images improve detection of panNENs [41,48]. DWI is also useful for metastasis and peritoneal carcinomatosis detection [21,49,50].

Pancreatic disease processes that should be considered in panNENs differential diagnosis on MRI include benign lesions such as intra-pancreatic accessory spleen, which typically shows lower ADC values, along with malignant processes, such as solid-appearing serous cystic neoplasms and pancreatic metastasis [51,52].

Furthermore, MRI may have a potential role in the surveillance of patients with MEN-1, in order to reduce radiation dose exposure [53].

## 3. Functional Imaging

Nuclear medicine has identified, as functional imaging’s target for gepNENs, the somatostatin receptors (SSTRs), which are expressed in 50–80% of panNENs [54].

However, SSRTs expression is not specific, because SSTRs are expressed by brain, pituitary, gastrointestinal tract, pancreas, thyroid, spleen, kidney, immune cells, vessels and peripheral nervous system [55]. Moreover, a wide variety of tumors show SSTRs expression, such as bronchopulmonary carcinoids, pituitary adenoma, pheochromocytoma, paraganglioma, neuroblastoma, medullary thyroid cancer and small-cell lung carcinoma [56].

Nuclear imaging is commonly indicated to perform staging, localization of a primary unknown NEN in patients with proved neuroendocrine metastasis, determination of radiotracer uptake for radionuclide therapy management and post-therapeutic re-staging [57].

Radiotracers consist of somatostatin analogues (SSAs); among somatostatin-receptor family members, SSAs have a high affinity only for subtypes sst2, sst3 and sst5 [21].

First molecular method to be introduced was Somatostatin Receptor Scintigraphy (SRS) with 111-pentetrotide (OCTREOSCAN; Mallinckrodt, St Louis, MO, USA) to date the most used and available SSA tracer.

More recently, the Food and Drug Administration approved 68-Gallio labeled SSA, allowing the use of positron emission tomography (PET) as functional imaging of choice due to its superior spatial resolution. In addition, PET/CT has the possibility of to perform a metabolic nuclear imaging through the use of 18F-fluorodeoxyglucose (18F-FDG) or 18F-L-dihydroxyphenylalanine (18F-DOPA). To date, guidelines indicate Octreoscan as a receptor tracer in gepNEN, but it will probably be replaced by Ga68-SSA tracers, due to their greater sensitivity for both primary lesion and metastasis [58]. Moreover, 68-Ga-DOTA-Exendin-4 PET/CT showed 94% accuracy in insulinomas localization [59], suggesting a role when preoperative localization of insulinomas fails with conventional imaging [60]. More recently, 68-Ga PET/MRI has been proposed as an alternative to PET/CT [61].

Peptide receptor radiometabolic treatment (PRRT) is recognized as a second- or third-line medical alternative by the recently published ENETS guidelines for the management of metastatic NEN [62]. Therefore, the individualization of the best candidates for PRRT is urgently needed. Some studies suggest that gepNENs with high uptake at SRS can be considered candidates for PRRT [63].

### 3.1. SRS

Scintigraphy protocol consists in a whole-body 2D (anterior–posterior) acquisition at 24h; 3D images can be performed with single photon emission computed tomography (SPECT), using the same radiotracer. This functional technique is highly specific, with a specificity value range between 92 and 100% [21]. The sensitivity value ranges between 40 and 70% [63] and is greater for well-differentiated gastrinoma, glucagonoma, VIPoma and non-functioning panNEN, while its value decreases in case of poorly differentiated NEC (Figure 2), insulinoma or small-size NEN [64,65].

### 3.2. PET/CT

PET/CT with 68-Gallio labeled SSA is the imaging method of choice for panNEN functional study. There are mainly three available SSAs showing no substantial differences in patients staging: 68-Ga-DOTANOC, 68-Ga-DOTATOC (Figure 5) and 68-Ga-DOTATATE. All of them are SSRT2 high-affinity tracers; only 68-Ga-DOTANOC proved to have an affinity for SSRT3.

Compared to SRS, PET/CT has shown higher sensitivity and specificity for identifying receptor-positive panNENs [66]. Several studies on the diagnostic accuracy of 68-GaSSTA PET in gepNENs have demonstrated a sensitivity range of 88–93%, and specificity range of 88–95% [21]. PET/CT and CT should be considered as complimentary procedures for patients with a suspected panNEN. Indeed, Versari et al. calculated a CT sensitivity of 91% and a mean 92% with 68-Ga DOTA-TOC for detection of duodeno-pancreatic neoplasia [67].

Furthermore, this nuclear technique has other advantages: the study has a lower radiation dose for the patient, thanks to the 2 h protocol, in comparison to 24 h of SRS; 68-GA SSA tracers have a favorable normal biodistribution, with a lower liver up-take, which can increase metastasis detection rate; background physiological uptake is similar to Octreoscan, with prominent uptake in adrenal glands and in the uncinated process of the pancreas, so that can be misdiagnosed as panNEN [68]. Standardized uptake values (SUVs) are calculated for quantitative analysis, allowing a direct comparison among exams performed with the same radiotracer.

Other positron-emitting biomarkers, such as 18-FDG and 18-F DOPA, are used to assess metabolic imaging, but their role is still unclear [69]. Moreover, 18-FDG PET might be useful for detecting aggressive neoplasms with increased levels of Ki67 and low SSTRs expression (Figure 2) [70]. Furthermore, 18-FDG uptake seems to be negatively correlated with prognosis [71]. Infectious and inflammatory processes can cause false-positive interpretations.

PET/CT plays a crucial role in metastasis detection, especially bone ones, when CT is negative or ambiguous. In a study by Prasad et al., 68-Ga-DOTANOC PET helped identify lymph node involvement in all patients, whereas CT identified nodal involvement in only 50% of patients [72].

Future improvement of nuclear imaging may consider dual-tracer PET/CT with both 18F-FDG and 68-Ga-DOTATATE to better differentiate G3 NETs, which show increased uptake with 68-Ga-DOTATATE, from panNECs, avid of 18F-FDG PET/CT [73].

## 4. Imaging Prognostic Factors

Since prognosis and treatment planning of panNENs are based on histologic differentiation and grading, many studies researched a relationship between CT/MRI findings and tumor grade, with most of them based on the 2010 WHO classification [74,75,76].

However, the role of CT in differentiation of panNENs grades is still limited. In a retrospective study by Zamboni et al. on 148 histologically confirmed panNENs, none of the analyzed parameters turned out to be significant predictors of G1/G2 tumors. The combination of the parameters with better performance (hypervascularity in the arterial phase, hyperdensity in the venous phase and well-defined margins) in recognizing G1/G2 tumors provided a low sensitivity (47%), higher specificity and positive predictive value (88% and 97%, respectively). On the other hand, G3 tumors resulted in being larger and being more often non-hypervascular in the arterial phase as compared to G1/G2 tumors [77]. Accordingly, in a previous work by Takumi et al., G2 tumors were found to be significantly larger in tumor size than G1 tumors (*p* = 0.029) [78]. In their study on 25 patients with panNENs, Zhu et al. found that, among the features of preoperative MDCT, lymphadenopathy and peripancreatic fat or vascular invasion were inclined to higher histopathological grading (tending to be malignant) [79].

More prognostic information can be obtained with MRI. Many studies found MRI morphological features of aggressiveness, such as size greater than 2 cm, irregular margins and atypical vascular appearance [46]. In addition, a correlation between DWI signal intensity, ADC value and panNENs grading has been widely suggested. Guo et al. observed a sensitivity and specificity of 72.3% and 91.6%, respectively, using an ADC value cutoff of 0.95 × 10^−3^ mm^2^/s for differentiating panNET G3 from panNETs G1/G2 [47]. In another more recent study, Guo et al. corroborated the inverse correlation between ADC values and tumor grade, demonstrating that ADC mean values was higher in well-differentiated G3 tumors (0.97 ± 0.16 10–3 mm^2^/s) than in poorly differentiated G3 tumors (0.69 ± 0.19 mm^2^/s) [80]. Lotfalizadeh et al. confirmed the prognostic usefulness of ADC values and the inverse correlation with Ki-67 [81]. Recently, in a retrospective analysis focused on predicting tumor grade with whole-tumor histogram analysis of ADC maps, De Robertis et al. found out that ADC_entropy_ is significantly higher in G2/G3 tumors (95% CI: 36.1–81.7). ADC_kurtosis_ was higher in panNENs with vascular involvement, nodal and hepatic metastases [82]. A large meta-analysis showed a pooled sensitivity and specificity for ADC in distinguishing G3 from G1/G2 of 0.93 (95% CI, 0.66–0.99) and 0.92 (95% CI, 0.86–0.95), respectively [83]. Moreover, tumor size, enhancement pattern and ADC value are independent prognostic factors for recurrence after curative resection of panNENs [84].

Some studies showed that SUV max values on 68 Ga-DOTATATE seem to have an inverse relationship with Ki-67. Both FDG PET/CT and 68 Ga-DOTATATE had a higher sensitivity for well-differentiated G2 and G3 tumors than for poorly differentiated G3 tumors [70,85].

Traditional diagnostic imaging has limitations for preoperative prognostication due to an evaluation of subjective qualitative imaging features. On the other hand, radiomics allows for the extraction of more objective quantitative features “hidden” inside the radiological images, using advanced texture and shape analysis [86,87,88,89]. With the advent of radiomics, many studies have tried to identify preoperative radiomics features able to predict tumor grade using CT and MR [90,91,92,93]. Good results have been achieved in terms of grading classification by radiomics in clear cell renal cell carcinoma, colorectal adenocarcinoma and gliomas [94,95,96,97]. Recent studies have focused the research on panNENs by using the radiomic method [97]. On CT, quantitative radiomics features, such as entropy, uniformity and kurtosis, are significantly different between well-differentiated G1/G2 tumors and poorly differentiated G3 tumors. In particular, entropy is the feature with the highest sensitivity in differentiating panNEN tumor grades [98]. Even if the research in this field is still limited, some studies build a radiomic-based predictive model to preoperatively and noninvasively differentiate tumor grades in patients with panNENs. Liang et al., in their study on 137 patients diagnosed with panNENs, found 233 radiomics features with statistical significance (*p* < 0.01) between the G1 and G2/G3 groups extracted from contrast-enhanced CT images. In addition, their nomogram developed by combining the radiomics signature with clinical stage showed a favorable result in predicting the histologic grade (G1 vs. G2/G3) with an AUC of 0.894 [99]. The multicenter study by Gu et al. on 138 patients with pathologically confirmed panNEN, building a comprehensive model consisting of tumor margin and fusion radiomic signature from the arterial and portal venous phase CT images for the preoperative prediction of histologic grade of panNENs (G1 vs. G2/G3): their nomogram obtained a good performance with AUC 0.974 and 0.902 in the training and validation cohorts, respectively [80]. On MRI, tumor texture analysis performed to predict tumor grade found that skewness and kurtosis values increase as tumor grade increases [100].

## 5. Conclusions

PanNENs are rare neoplasms that require a multidisciplinary and multimodal approach. Morphological imaging (CT, MRI and US) and functional imaging (111In-Octreoscan, 68-Ga-DOTATATE,18F-FDG PET, etc.) play a complementary role for panNEN detection, staging and surveillance. Moreover, the future challenge of imaging is to provide preoperative prognostic information. Promising results in this field have been achieved by radiomics. Knowledge of recent updates in imaging modalities and familiarity with typical/atypical imaging findings of panNENs are essential for the management of these patients.

## Figures and Tables

**Figure 1 ijerph-18-08895-f001:**
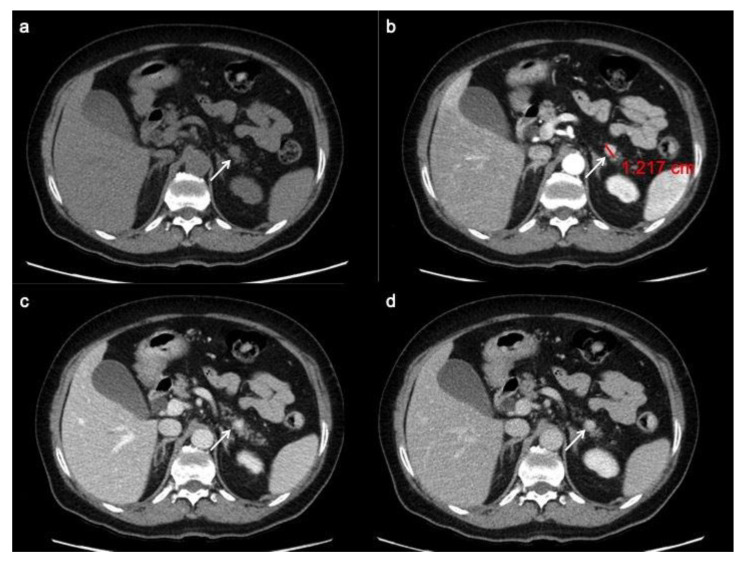
PanNET G1 according to the 2019 WHO classification. CT images in the transverse plane during the basal (**a**), the arterial (**b**), the portal venous (**c**) and the delay (**d**) phases show a small pancreatic hypervascularized tumor of the pancreatic tail with sharp margin (arrow).

**Figure 2 ijerph-18-08895-f002:**
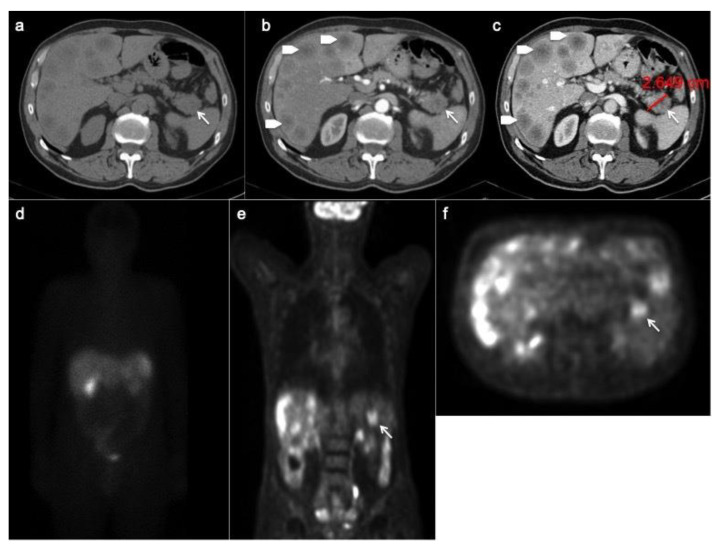
Patient with liver metastases from poorly differentiated pancreatic neuroendocrine carcinoma. CT images in the transverse plane during the basal (**a**), the arterial (**b**) and portal (**c**) phases show a large mass developed in the pancreatic tail with atypical hypo-enhancing pattern (arrow). This lesion is associated with multiple liver metastases (arrowheads) that appear hypodense with rim enhancement during arterial phase (**b**). According to the poorly differentiated tumor feature, FDG PET/CT (**e**,**f**) shows high uptake in pancreatic lesion while Somatostatin Receptor Scintigraphy with ^111^-pentetrotide is negative (**d**).

**Figure 3 ijerph-18-08895-f003:**
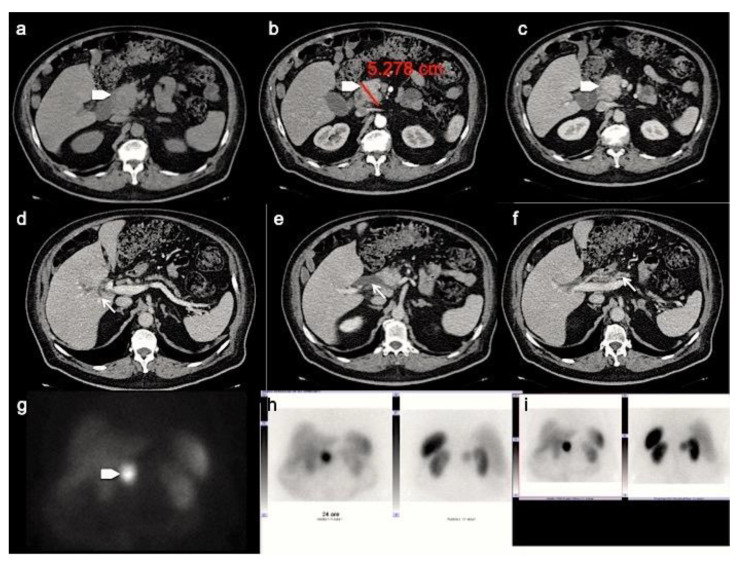
CT images in the transverse plane during the basal (**a**), the arterial (**b**) and portal (**c**) phases show a large mass developed between the second portion of the duodenum and the pancreatic head (arrowhead). This panNEN is associated with a dilation of the intrahepatic biliary tree (**d**), common bile duct (**e**) and main pancreatic ductal (**f**) (arrows). Somatostatin Receptor Scintigraphy with ^111^-pentetrotide shows high uptake of somatostatin analogue in pancreatic lesion (**g**–**i**).

**Figure 4 ijerph-18-08895-f004:**
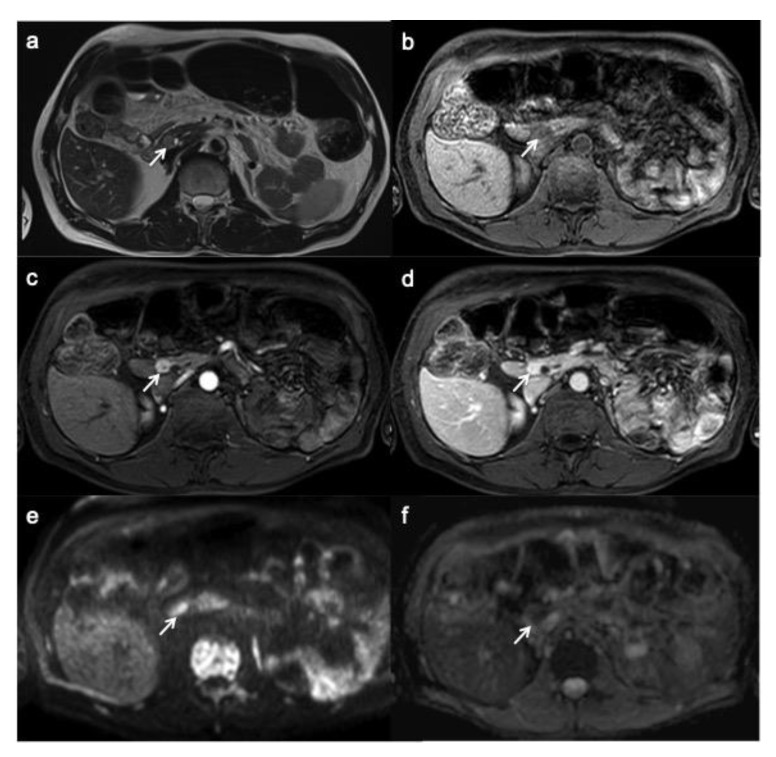
Small functioning pancreatic neuroendocrine tumor. T2-weighted MR image (**a**) shows a small lesion with well-circumscribed margin and high signal intensity (arrow). On T1W image, the lesion appears hypo-intense within the surrounding hyperintense pancreatic parenchyma (**b**). On contrast enhancement sequences during the arterial (**c**) and portal (**d**) phases, the lesion shows hyper-enhancement (arrow). On DWI (**e**) and ADC map (**f**), the lesion shows a clearly restrictive pattern.

**Figure 5 ijerph-18-08895-f005:**
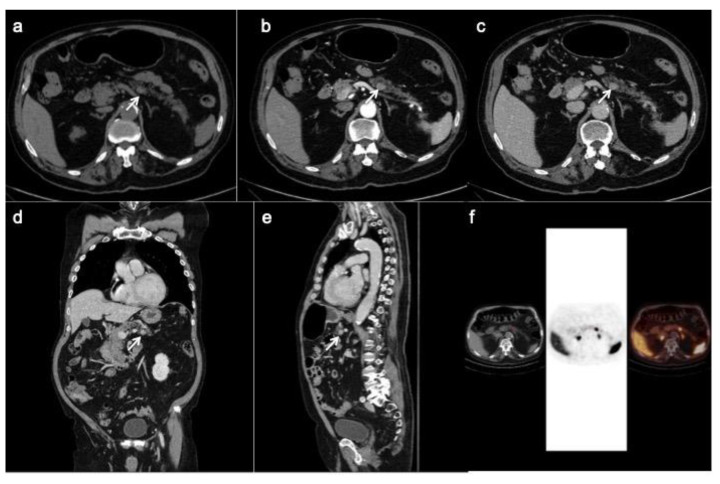
CT images (**a**–**c**) show a hyper-enhancing pancreatic nodule in the body/tail of the organ (arrows) well depicted on the coronal (**d**) and sagittal (**e**) reconstruction too. The 68-Ga-DOTATOC PET/CT (**f**) shows focal uptake in the pancreatic lesion.

## Data Availability

Not applicable.

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
