# Peer review of "Imaging of Pancreatic Neuroendocrine Neoplasms"

_ijerph, 2021, doi:10.3390/ijerph18178895_

Round 1

Reviewer 1 Report

In this manuscript, researchers have demonstrated imaging of pancreatic neuroendocrine neoplasms, so called panNENs). CT and MRI imaging modalities have been used for tumor detection. The research on pancreatic disease has potential and contribute to the understanding of the mechanism. The writing has been prepared well. The literature search is sufficient. Minor comments will on the figure. Could research add scale bar? Overall, the quality of this manuscript is meeting the basic requirement of this journal. "Accepted" is suggested.

Author Response

Minor comments will on the figure. Could research add scale bar? 

I could not add scale bar but some sizes of the pancreatic lesions have been added

Reviewer 2 Report

This paper introduces a review recent publications
for the topic of Imaging of pancreatic neuroendocrine neoplasms.
The topic selected by the authors is interesting from several
points of view, for instance research, applications and to
improve life expectancy.
In a widespread way,
the paper is well written  paper: in methodology, 
documentation of sources, references and organization.
However, some issues to  
improved are the following:

(1) The abstract is an introduction to the importance of the topic.
 However, it does not summarize the article. Then, it needs to  
 be completely rewritten according to the standard requirements of 
 an abstract.

(2) A paragraph reviewing some similar works or related works 
is missing in the introduction.
Then, it should be included.

(3) There is not a conclusions section. Then a new section
of conclusions with possible challenges for future work should be included.

(4) Some strange symbols appear in different parts of the text, 
even in the bibliography (see the marks on the pdf file).

Finally, my conclusion is the following: ``
The results obtained in the paper interesting. 
However, some minor inaccuracies are necessary to be fixed. 
After the revision I will have 
no objection to recommend it for
publication in IJERPH.''

Author Response

(1) The abstract is an introduction to the importance of the topic.
 However, it does not summarize the article. Then, it needs to  
 be completely rewritten according to the standard requirements of 
 an abstract.

The abstract was modified according to the reviewer’s comments.

(2) A paragraph reviewing some similar works or related works 
is missing in the introduction.
Then, it should be included.

The paragraph reviewing some similar works or related works has been added in the introduction as suggested by the reviewer.

(3) There is not a conclusions section. Then a new section
of conclusions with possible challenges for future work should be included.

Conclusions have been added as suggested by the reviewer.

(4) Some strange symbols appear in different parts of the text, 
even in the bibliography (see the marks on the pdf file).

I removed the strange symbols appearing in the text.

Round 2

Reviewer 2 Report

In my opinion, the article is interesting and would be possible to publish in the present form.